# Genome-Wide Associations and Confirmatory Meta-Analyses in Diabetic Retinopathy

**DOI:** 10.3390/genes14030653

**Published:** 2023-03-05

**Authors:** Xinting Yu, Shisong Rong

**Affiliations:** 1Department of Medicine, Brigham and Women’s Hospital, Mass General Brigham, Harvard Medical School, Boston, MA 02445, USA; 2Department of Ophthalmology, Massachusetts Eye and Ear, Mass General Brigham, Harvard Medical School, Boston, MA 02445, USA

**Keywords:** diabetic retinopathy, genome-wide association study, genetic associations, systematic review, meta-analysis, diabetes mellitus, diabetic vasculopathy, functional annotation

## Abstract

The present study aimed to summarize and validate the genomic association signals for diabetic retinopathy (DR), proliferative DR, and diabetic macular edema/diabetic maculopathy. A systematic search of the genome-wide association study (GWAS) catalog and PubMed/MELINE databases was conducted to curate a comprehensive list of significant GWAS discoveries. The top signals were then subjected to meta-analysis using established protocols. The results indicate the need for improved consensus among DR GWASs, highlighting the importance of validation efforts. A subsequent meta-analysis confirmed the association of two SNPs, rs4462262 (*ZWINT-MRPS35P3*) (odds ratio = 1.38, *p* = 0.001) and rs7903146 (*TCF7L2*) (odd ratio = 1.30, *p* < 0.001), with DR in independent populations, strengthening the evidence of their true association. We also compiled a list of candidate SNPs for further validation. This study highlights the importance of consistent validation and replication efforts in the field of DR genetics. The two identified gene loci warrant further functional investigation to understand their role in DR pathogenesis.

## 1. Introduction

Diabetic retinopathy (DR) is a leading cause of visual impairment. It leads to blinding complications and affects more than one-third of global diabetic patients, posing blindness to one in three DR patients [1]. DR is a progressive retinal microvascular complication of diabetes mellitus. Its pathology involves microaneurysms, hemostatic changes, and capillary occlusion in the retina. The vessel permeability increases, resulting in retinal nonperfusion, neovascularization, and vessel wall thickening. Further progress into macular edema, vitreous hemorrhage, or retina distortion can lead to severe visual loss and blindness [2]. Effective interventions, such as tight glycemic and blood pressure control, lipid-lowering, and laser therapies, could delay its progression and preserve vision. Early diagnosis and appropriate treatment can help prevent visual damage and the blinding consequences [3,4,5,6,7,8].

Genetic studies play an important role in uncovering DR-related genes and shedding light on the underlying pathophysiology and potential diagnostic and therapeutic markers. Over 300 DR candidate gene association studies have been published [9]. Common variants within genes involved in various processes, such as angiogenesis, glucose metabolism, lipid metabolism, inflammation, vascular regulation, cell communication, and extracellular matrix, have been evaluated for their associations with DR and its subtypes, with noticeable inconsistent findings [10,11]. In an effort to address inconsistencies and uncover a broader range of genetic factors, the field has shifted towards genome-wide association studies (GWASs) for investigating DR.

Genome-wide association studies exam the entire genome in populations to identify single nucleotide polymorphisms (SNPs) that are associated with a particular trait or disease. GWASs have been conducted in different populations to investigate the genetic architecture of DR [12,13,14,15,16,17,18,19,20] and its subtypes [21,22,23,24,25,26,27,28], such as proliferative diabetic retinopathy (PDR) and diabetic macular edema (DME)/diabetic maculopathy. These GWASs have reported more than 100 gene loci and SNPs that significantly or suggestively alter the risk of DR in European, Asian, and African populations. However, the reported SNPs varied among the DR GWASs [10,11,12,13,29]. Therefore, a systematic summary and evaluation of existing DR GWAS results is needed to prioritize the efforts in replicating these GWAS signals in independent populations, ultimately advancing our understanding about the DR genetics.

In this study, we first summarized all reported SNPs from published GWASs on DR and its subtypes. Second, we sifted through SNPs of genome-wide or suggestive significance to identify overlapping signals between independent GWASs. Finally, we calculated the summary outcomes for the top GWAS SNPs using a meta-analysis and all available genetic data from independent replication studies. Our study provides a comprehensive summary and update, emphasizing the urgent need for improved consensus among DR GWASs. Through a meta-analysis of the reported GWAS SNPs, we confirmed the association of two SNPs, rs4462262 (*ZWINT-MRPS35P3*) and rs7903146 (*TCF7L2*), with DR. However, to ensure the robustness and generalizability of GWAS findings, further replication of the candidate SNPs summarized in our study is necessary.

## 2. Materials and Methods

### 2.1. Identification of DR GWASs and Data Extraction

Genome-wide association studies on DR were identified via searching the GWAS catalog [30], literature databases, and citation lists of relevant publications. The search strategies used for the identification of DR GWASs in PubMed/MEDLINE are detailed in Appendix A. The search terms used in the GWAS catalog were DR (trait ID in the Experimental Factor Ontology (EFO): EFO_0003770), PDR (EFO_0009322), DME (EFO_0009321), and diabetic maculopathy (EFO_0010133). We summarized all records that met the following criteria: (1) studies tested the associations of genetic variants with DR and its subtypes (i.e., nonproliferative DR (also known as NPDR), PDR, sight-threatening DR, DME, or diabetic maculopathy) at a genomic scale; (2) study population was clearly defined; (3) background disease was type 1 diabetes (T1D), type 2 diabetes (T2D), or both T1D and T2D; (4) diagnosis of DR was based on clinical data; and (5) publicly reported genome-wide significant variants.

The data extraction process for the SNPs associated with DR involved screening and extracting data from both GWAS catalog datasets and identified GWASs from a literature search. Two authors, X.Y.T. and S.S.R., reviewed the records, performed the data extraction, and resolved any discrepancies through consensus. The extracted data were then organized into summary tables for further analysis.

### 2.2. Identification of Overlapping Signals between GWASs

To identify overlapping SNPs between documented DR GWASs, we took a multistep approach. Three DR phenotypes were analyzed separately, including DR, PDR/sight-threatening DR, and DME/diabetic maculopathy. Within each phenotype group, the first step was to search for identical SNPs present in two or more independent DR GWASs. Secondly, for each DR phenotype, we searched for SNPs that were near each other within a ±100 K base pair window. This step aimed to identify SNPs that may be in linkage disequilibrium and may have a similar effect on the DR phenotypes. This method of searching for identical and nearby SNPs between different DR GWASs allowed for the identification of SNPs with the best available genetic evidence supporting their effects on the risk of DR and provided a list of SNPs for the subsequent meta-analysis.

### 2.3. Genetic Meta-Analysis

A genetic meta-analysis was conducted for any SNPs or gene loci repeatedly identified by independent DR GWASs and SNPs with a genome-wide significant *p*-value (<1 × 10^−7^) following previously published protocols [31,32,33,34,35]. Briefly, we performed the literature search using Boolean logic and search terms with controlled vocabularies (i.e., Medical Subject Heading terms) in the PubMed/MEDLINE databases. The search terms were constructed as follows: (Identified genes and genetic loci OR SNP IDs) AND (“diabetic retinopathy” (MeSH Terms) OR (“diabetic” (All Fields) AND “retinopathy” (All Fields)) OR “diabetic retinopathy” (All Fields)) (Appendix A). Additionally, we scanned the reference lists of relevant research articles, reviews, and meta-analyses identified during the screening process to include all relevant genetic data. The last search was conducted on 28 January 2023.

We included SNPs and studies that met the following criteria in the genetic meta-analysis: (1) original genetic case-control studies that enrolled unrelated individuals from explicitly defined populations; (2) studies that used T1D, T2D, or both as the background conditions; (3) odds ratio and 95% confidence intervals (CIs)/standard error (SE) were reported or calculable based on the reported data; and (4) SNPs with genome-wide significant *p*-values (<1 × 10^−7^) in the documented GWAS discovery cohorts, or the gene loci contained genome-wide significant SNPs. We excluded animal studies, case reports, reviews, abstracts, conference proceedings, and editorials. Two investigators (X.T.Y. and S.S.R.) screened, reviewed, and extracted all the data independently. Disagreements were resolved through consensus. If the allele counts were not provided, we calculated them from the genotype data. If only the OR and 95% CI were reported, we estimated the SE using the equation SE = (β−ln(lower limit of 95% CI))/1.96, where β = ln (OR) [36]. Fully adjusted outcomes were used in the meta-analysis when available. If duplicated cohorts were identified, we used the larger and more recent cohort in the meta-analysis. We also adopted the Newcastle Ottawa Scale (NOS) (accessed via http://www.ohri.ca/programs/clinical_epidemiology/oxford.asp, accessed on 1 January 2023) to evaluate the overall quality of the case-control studies (Appendix B) [37,38,39]. A study with ≤6 stars was considered high risk for introducing biases; therefore, it was subject to removal in the sensitivity analysis [40].

We performed a meta-analysis for each SNP if it was reported in two or more independent cohorts. The genetic association was evaluated using an allelic model, i.e., effect allele frequency vs. reference allele frequency. We calculated the summary OR and 95% CI for each polymorphism using the random-effects model. The heterogeneity was tested using the I^2^ statistics [41]. An I^2^ value lower than 50% indicated low heterogeneity. We plotted the funnel plots to assess the publication bias [42,43,44]. We also conducted a sensitivity analysis to confirm the associations by sequentially omitting each of the studies one at a time if the studies deviated from the HWE or were of suboptimal quality [31,45]. We performed the meta-analysis using RevMan 5 (https://training.cochrane.org/online-learning/core-software/revman/, accessed on 1 January 2023). Summary *p*-values of <0.05 were considered statistically significant.

### 2.4. Functional Annotation of SNPs and Gene Loci

To understand the functional relevance of the identified SNPs related to DR, we employed in silico functional prediction scores, such as SIFT [46], PolyPhen [47], CADD [48], and RegulomeDB [49]. Moreover, we also assessed the SNPs in high (D-prime ≥ 0.8 and r^2^ ≥ 0.8) linkage disequilibrium (LD) with the identified SNPs. The LD data for European populations from the 1000 Genome project were used [50]. Additionally, we analyzed the expression quantitative trait loci (eQTL) databases through the Genotype-Tissue Expression (GTEx) portal to gain direct knowledge of the effects of risk alleles on nearby gene expression [51].

## 3. Results

We identified 14 GWASs conducted in diverse populations, including European [12,21,23,24,25,27], Asian [12,21,23,24,25,27], African [26,28], and Arabian [15], through searching the GWAS catalog and literature databases (Appendix A and Table 1). With the exception of one GWAS that focused on T1D [21] and another GWAS that included both T1D and T2D [15], the majority of the GWASs enrolled individuals with DR in a T2D context. Only half of the GWASs tested significant SNPs discovered in the initial stage in at least one replication cohort [14,17,22,23,25,26,28]. Although the majority of GWASs had small sample sizes, with less than 1000 participants, no power analysis results were reported (Table 1). Additionally, we found three pertinent GWASs that used samples from the UK Biobank [18,19,20]. However, we excluded them from the summary table, as they primarily used nondiabetic population controls.

It is worth noting that there have been exciting developments in the field of DR genetics in recent years. These advancements include exome sequencing for rare and functional variants [15], and a more in-depth analysis of DR GWAS datasets has provided further understanding of the underlying pathological pathways [52], causal factors for DR [53,54,55], ability to predict DR using polygenic risk scores [56], and pharmacogenetic responses to treatment [57].

### 3.1. Genome-Wide Associations of DR

A total of six GWASs identified 76 SNPs located in 61 gene loci, with *p*-values that suggest a potential or confirmed genome-wide significance for DR (*p* < 5 × 10^−5^) (Appendix A) [13,14,16,17,21,26]. Our subsequent analysis revealed a lack of consensus among the existing DR GWASs, as we found no overlap in SNPs from different studies within a ±100 K base pair window.

Seven out of the seventy-six SNPs exhibited genome-wide significance (*p* < 1.0 × 10^−7^), with the top seven being rs17376456 (*KIAA0825/C5orf36*, *p* = 3.0 × 10^−15^), rs2038823 (*HS6ST3*, *p* = 5.0 × 10^−11^), rs12630354 (*THRAP3P1-STT3B*, *p* = 7.0 × 10^−10^), rs4838605 (*ARHGAP22*, *p* = 2.0 × 10^−9^), rs12219125 (*AMD1P1-PLXDC2*, *p* = 9.0 × 10^−9^), rs202069793 (*OR13D3P-OR13D1*, *p* = 6.0 × 10^−8^), and rs4462262 (*ZWINT-MRPS35P3*, *p* = 9.0 × 10^−8^) (Appendix A). Notably, two of these seven SNPs, rs12630354 and rs202069793, were validated in separate populations in the original GWASs [17,26]. All seven SNPs were included in subsequent meta-analysis.

### 3.2. Genome-Wide Associations of PDR and DME

In our investigation of PDR, we identified six GWASs that tested PDR against a T2D background (Appendix A) [22,23,24,25,26,28], with all but one study including a validation cohort [24] (Table 1). A total of 29 SNPs located near 27 gene loci demonstrated suggestive or lower *p* values. Among them, eight SNPs showed genomic significance, including rs3081219 (*WDR72*, *p* = 1.0×10^−9^), rs3913535 (*NOX4*, *p* = 4.0 × 10^−9^), rs11018670 (*FOLH1B*, *p* = 1.0 × 10^−8^), rs72740408 (*HNRNPA1P46*, *p* = 2.0 × 10^−8^), rs184340784 (*LINC01646*, *p* = 4.0 × 10^−8^), rs1065386 (*HLA-B*, *p* = 5.0 × 10^−8^), rs4726066 (*PRKAG2*, *p* = 5.0 × 10^−8^), and rs200295620 (*GOLIM4-EGFEM1P*, *p* = 7.0 × 10^−8^).

Two GWASs were identified for DME, which collectively found 12 SNPs located near 10 gene loci (Appendix A) [24,27]. Notably, only one of the SNPs, rs9966620, was genome-wide significant (*p* = 7.0 × 10^−8^). Neither of the two GWASs included a validation cohort (Table 1).

Furthermore, our analysis revealed a lack of overlapping SNPs or gene loci among different GWASs, and neither PDR nor DME had overlapping SNPs or gene loci among the different GWASs within a ±100 K base pair window, which was comparable to the results obtained from the DR GWASs.

### 3.3. Genetic Meta-Analysis of Top GWAS Signals

#### 3.3.1. Meta-Analysis of Top DR-Associated SNPs

Replication studies and meta-analyses are crucial for validating findings and strengthening evidence for a true association, especially given the lack of consensus among existing GWASs. In our meta-analysis, we identified 37 records of published studies from the literature search and extracted data from 20 replication studies (Appendix A). Four of the seven top DR GWAS SNPs (Appendix A) were replicated in two or more independent studies, including rs4462262 (*ZWINT-MRPS35P3*) [58,59,60], rs12219125 (*PLXDC2-NEBL*) [59,60,61], rs4838605 (*ARHGAP22*) [58,59,60], and rs17376456 (*C5orf36*) [59,60]. Notably, all four of these SNPs were reported without an internal validation cohort in the initial GWAS [13]. The remaining three of the seven GWAS SNPs, rs2038823 [13], rs12630354 [17], and rs202069793 [26], have not been replicated in independent studies. Additionally, we found five replication studies [62,63,64,65,66] that tested the DR-association of rs7903146 in the *TCF7L2* gene locus, which was identified by rs34872471 (*p* = 4 × 10^−15^) in a UK Biobank GWAS using population controls [20]. These two SNPs were in high linkage disequilibrium in European populations (r^2^ ≥ 0.99) [50]. Therefore, we conducted a meta-analysis for rs7903146.

Our meta-analysis confirmed the significant association of rs4462262 (OR = 1.38, *p* = 0.001), rs12219125 (OR = 1.24, *p* = 0.03), and rs7903146 (OR = 1.30, *p* < 0.001) with DR in replication cohorts (Figure 1 and Appendix A). The heterogeneity measurements (I^2^) were low for all of these combined outcomes. However, we found no evidence of association between rs4838605 and rs17376456 with DR in our analysis (*p* > 0.05) (Figure 1). 

All studies included in the meta-analysis scored six or higher on the NOS quality assessments (Appendix A). Funnel plots showed no significant deviations (Appendix A), indicating a lower risk of introducing potential biases by pooling results from these studies. The sensitivity analysis confirmed the stability of most of the meta-analytical results by omitting one study at a time and recalculating the combined outcomes. Only rs12219125 became insignificant after removing Hosseini S. M.’s study [61], highlighting the need for further replication efforts.

#### 3.3.2. Meta-Analysis of Top PDR- and DME-Associated SNPs 

Our literature search did not identify replication efforts for the top GWAS hits of PDR and DME, preventing us from performing meta-analyses for these DR phenotypes.

### 3.4. Biological Relevance of rs4462262 and rs7903146 to DR

The SNP rs4462262 is an intergenic variant, located approximately 750 kbp upstream to the nearest gene, *IPMK*. A microRNA, MIR3924, was situated approximately 110 kbp upstream of the SNP. However, the functional annotations for rs4462262 and 54 other SNPs in high LD with rs4462262 were unremarkable (Table 2), providing limited evidence for the biological relevance of these variants.

In contrast, the SNP rs7903146 is located in intron 3 of *TCF7L2* and is significantly associated with the expression of *TCF7L2* (*p* = 2.9 × 10^−7^) (Table 2). Another SNP, rs7074440, in high LD with rs7903146 (D’ = 0.95 and r^2^ = 0.91) has a CADD score of 10.1 and is also significantly associated with the expression of *TCF7L2* (*p* = 2.1 × 10^−6^) (Table 2), indicating a potentially functional role in *TCF7L2* regulation.

## 4. Discussion

In this study, we comprehensively summarized the genomic association signals of DR, PDR, and DME/diabetic maculopathy from published GWASs. Our results emphasized the urgent need for improved consensus among DR GWASs, underlining the importance of validation and replication efforts. We confirmed the association of two SNPs, rs4462262 (*ZWINT-MRPS35P3*) and rs7903146 (*TCF7L2*), with DR through meta-analysis in independent populations, strengthening the evidence of their true association. Moreover, we provided a list of candidate SNPs for further validation studies.

The lack of consensus among DR GWASs is a complex issue that arises from several potential factors. One such factor is the clinical heterogeneity of DR populations, which can lead to conflicting results between studies. A possible solution is to carefully stratify patients based on specific clinical characteristics and to conduct studies in homogeneous patient populations. Second, ethnic differences in SNP allele frequencies can impact the strength and direction of genetic associations. To address this issue, large-scale studies with diverse populations are needed to increase the generalizability of results. Additionally, the standardization of SNP definitions and frequency databases can improve the accuracy of genetic association analyses [67]. Third, the study definition of DR and its subtypes could be another factor contributing to the lack of consensus. To address this, a standardized definition and diagnostic criteria, such as the ETDRS grading system [68,69], should be universally adopted by the DR genetic research community to ensure consistency between studies. Fourth, whether to include a validation design within a GWAS is another potential factor. Including a validation design within a GWAS can increase the confidence in the findings and help to reduce false positive results. This is particularly important in a complex disease such as DR, where multiple genetic and environmental factors are likely to be involved [11,70]. Finally, differences in the statistical power between studies can also contribute to the lack of consensus among DR GWASs. To overcome this issue, larger sample sizes and more powerful study designs are needed to increase the chances of detecting true associations, especially in rare variant studies. Addressing these potential factors through careful study design and increased collaboration among DR researchers can help to improve the consistency of DR GWAS results and eventually lead to a better understanding of the underlying mechanisms of DR.

In our meta-analysis, we noticed a lack of replication studies for DR GWASs findings. Replication is a critical step to confirm the validity of a discovery and strengthens the evidence for a true association. Without replication, the results of a study may be considered unreliable or of limited generalizability, especially when the study is based on a small sample size or has limited statistical power. However, replication studies are often challenging and resource intensive, which can limit the number of independent validation studies performed for DR GAWS discoveries. This is particularly true in the case of DR genetics, which are often complex due to the heterogeneity of DR populations, differences in SNP allele frequencies across ethnic groups, and variations in the definition of DR and its subtypes. Therefore, it is essential to prioritize replication efforts in DR genetics research, which directly pointed us towards conducting this study.

The association between rs7903146 and DR is significant, and the regulatory effects of this SNP and other SNPs in LD provide evidence supporting the functional role of TCF7L2 in DR pathogenesis. As a transcriptional regulator of the canonical Wnt signaling pathway, *TCF7L2* plays a vital role in cell proliferation and fate specification. The involvement of *TCF7L2* in DR pathogenesis can be two-fold. First, TCF7L2 may promote pathological retinal neovascularization via the ER stress-dependent upregulation of VEGFA, making retinal cells from rs7903146-TT patients more sensitive to VEGF upregulation and at greater risk of developing PDR [63]. Second, *TCF7L2* may also increase the risk of DR development and progression by influencing the responses to glycemic control drugs, such as glipizide and metformin, which are two commonly used glycemic control drugs [71]. In addition, good glycemic control is closely associated with short- and long-term beneficial effects on DR prevention and intervention [72,73]. However, the functional relevance of rare mutations in the TCF7L2 gene and their impact on DR pathophysiology in diverse ethnic groups remain unknown, and further sequencing studies of large cohorts are needed to understand their role. In addition, understanding the functional regions of the TCF7L2 gene and their impact on specific phenotypes will also be important for comprehending their role in diverse human diseases, including DR [74,75].

On the other hand, the lack of functional annotations for rs4462262 and 54 other SNPs in high LD with rs4462262 limits our understanding of their functional relevance in DR pathogenesis. To reveal their role in DR, the identification of rare and common variants that truly alter the risk of DR and functional investigations are crucial. The confirmation and functional annotation of these variants could be a critical step in discovering new genetic markers for DR, facilitating better prediction of disease susceptibility and tailoring individualized treatments for patients.

This study has several potential limitations, including the limited number of independent replication cohorts and subphenotypes, limited diversity of population ancestries, and the use of population controls instead of DR-free diabetic controls in the DR GWASs using the UK Biobank datasets. These limitations affected our ability to confirm more GWAS discoveries.

## 5. Conclusions

Our study underlines the urgent need for consistent replication efforts in the field of DR genetics. Two SNPs, rs4462262 (*ZWINT-MRPS35P3*) and rs7903146 (*TCF7L2*), showed consistent association with DR. Confirmation and functional investigations of reported and new variants are needed for a better understanding of the underlying mechanisms of DR development.

## Figures and Tables

**Figure 1 genes-14-00653-f001:**
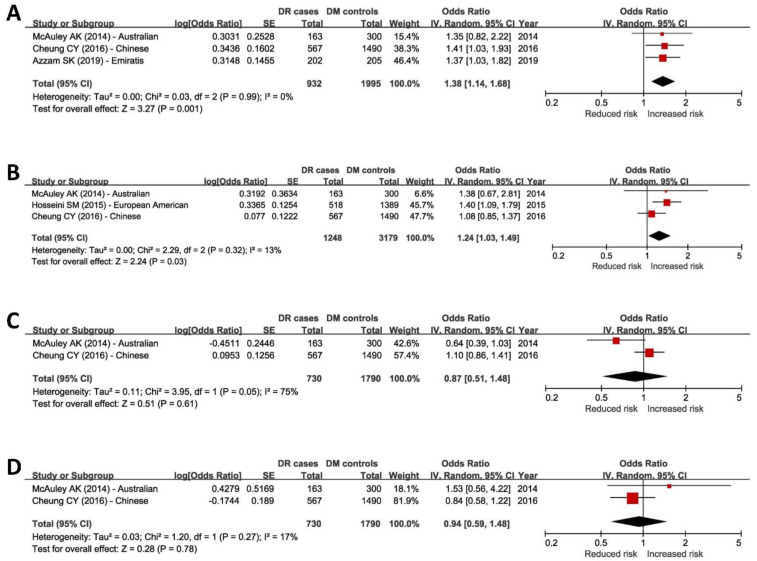
Combined replication outcomes of four genome-wide significant DR GWAS SNPs using meta-analysis A [58,59,60,61]. (**A**) The combined odds ratio suggested a significant association of the rs4462262 T allele with increased risk of DR (*p* = 0.001). The minimum heterogeneity was detected using I^2^. (**B**) The combined odds ratio suggested a significant association of the rs12219125 T allele with increased risk of DR (*p* = 0.03). The heterogeneity among the pooled studies was low. (**C**) The combined odds ratio did not support a significant association of rs12219125 with DR (*p* = 0.61). The heterogeneity among the pooled studies was high. (**D**) The combined odds ratio did not support a significant association of rs17376456 with DR (*p* = 0.78). The heterogeneity among the pooled studies was low. CI, confidence interval; DM, diabetes mellitus; DR, diabetic retinopathy; SE, standard error.

**Table 1 genes-14-00653-t001:** Genome-wide association studies of diabetic retinopathy.

	First Author	Year	Studied Trait	Background Disease	Discovery Population	Replication Population	Discovery Sample Size (N)			Significant SNP	Reference
							Case	Control	Total		
1	Fu, Y.P.	2010	DR	T2D	Hispanic	None	103	183	286	0	[12]
2	Huang, Y.C.	2011	DR	T2D	Chinese	None	174	575	749 *	8	[13]
3	Grassi, M.A.	2011	DR (DME + PDR)	T1D	European	None	973	1856	2829	19	[21]
4	Sheu, W.H.	2013	PDR	T2D	Chinese	Hispanic or Latin American	437	570	1007	1 **	[22]
5	Awata, T.	2014	DR	T2D	Japanese	Asian	205	241	446	1	[14]
6	Burdon, K.P.	2015	DR (severe NPDR + PDR + DME)	T2D	Australians (European)	European, Indian	336	508	844	1	[23]
7	Shtir, C.	2016	DR	DM	Saudi Arabian	None	43	64	107	3	[15]
8	Graham, P.S.	2018	DME and PDR	T2D	Australians (European)	None	270 and 176	435	881	2 and 2	[24]
9	Meng, W.	2018	Severe DR	T2D	Scottish (European)	Independent meta-analysis	560	4106	4666	2	[25]
10	Pollack, S.	2018	DR and PDR	T2D	African American, Afro-Caribbean, European	Asian, European, Hispanic	MDP	MDP	5857	19 and 20	[26]
11	Meng, W.	2019	Diabetic maculopathy	T2D	European	None	469	1374	1843	8	[27]
12	Liu, C.	2019	PDR	T2D	African	African American or Afro-Caribbean	64	227	291	4	[28]
13	Hsieh, A.R.	2020	DR	T2D	Chinese	None	206	206	412	3	[16]
14	Imamura, M.	2021	DR	T2D	Japanese	Japanese	4839	4041	8880	6	[17]

* Nondiabetic controls were included. ** Only significant in the discovery cohort. DME, diabetic macular edema; DM, diabetes mellitus; DR, diabetic retinopathy; SNP, single nucleotide polymorphism; T1D, type 1 diabetes; T2D, type 2 diabetes; MDP, more details in the publication; NPDR, nonproliferative diabetic retinopathy; NR, not reported; PDR, proliferative diabetic retinopathy.

**Table 2 genes-14-00653-t002:** Functional relevance of rs4462262 and rs7903146 to diabetic retinopathy.

	SNP	Position	Location	Gene	SIFT	PolyPhen	CADD	RegulomeDB *	eQTL (*p*)
1	rs4462262	Chr10:59189178	Intergenic	None	na	na	7.10	5	na
2	rs7903146 **	Chr10:114758349	Intronic	*TCF7L2*	na	na	3.27	5	2.9 × 10^−7^
3	rs7074440	Chr10:114785424	Intronic	*TCF7L2*	na	na	10.10	2b	2.1 × 10^−6^

* 2b, TF binding + any motif + DNase Footprint + DNase peak; 5, TF binding or DNase peak. The complete scoring scheme is available at https://www.regulomedb.org/regulome-help/ (accessed on 10 February 2023). ** In complete linkage disequilibrium with rs34872471 identified in GWAS by Xue, Z. et al. [20]. CADD, combined annotation-dependent depletion score; eQTL, expression quantitative trait locus; na, not available; PolyPhen, polymorphism phenotyping score; RegulomeDB, link: https://cherrylab.stanford.edu/projects/regulomedb (accessed on 10 February 2023); SIFT, sorting intolerant from tolerant score.

## Data Availability

We used publicly available software for the analyses and provided a list of the specific programs used in Section 2.

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
