# Peer review of "Genome-Wide Associations and Confirmatory Meta-Analyses in Diabetic Retinopathy"

_genes, 2023, doi:10.3390/genes14030653_

Round 1
Reviewer 1 Report
In this work, Yu et al. summarized the genomic association signals for DR and its subtypes by meta-analysis con-firmed the association of two SNPs, rs4462262 (ZWINT-MRPS35P3) and rs7903146 (TCF7L2). Although this work might be of interest, some concerns need to be addressed:
1. The authors only pointed out the possible association of rs4462262 (ZWINT-MRPS35P3) and rs7903146 (TCF7L2) with DR, which is not sufficient. In silico functional annotation or experimental validation are required to support the authors' claims. Moreover, possible association between these SNPs and DR pathogenesis should be covered in the Discussion.
2. The first two paragraphs of the Introduction both presented general knowledge of DR, which can be merged into one paragraph.
3. The authors should highlight the significance and novelty of their work in the last paragraph of the Introduction and the Discussion.
4. The hyphens in line 14 and 16 should be removed.
5. The column name “PMID” in Table 1 should be corrected.
Author Response
Dear Reviewer,
Thank you for taking the time to review our manuscript and for providing us with your valuable feedback. We are grateful for your input and your expertise in helping us improve the quality of our research.
We appreciate the opportunity to revise and improve our manuscript based on your suggestions. Per your valuable feedback, we have thoroughly revised the manuscript and included a point-by-point response to address all the issues and concerns raised in your review. Additionally, to make the changes more visible, we have highlighted them in yellow throughout the manuscript.
Once again, we appreciate your support and guidance throughout this process, and we hope that our revised manuscript meets your expectations.
Thank you for your valuable feedback!
Sincerely,
Shisong Rong
Department of Ophthalmology, Massachusetts Eye and Ear,
Mass General Brigham, Harvard Medical School,
Boston, MA, USA
In this work, Yu et al. summarized the genomic association signals for DR and its subtypes by meta-analysis confirmed the association of two SNPs, rs4462262 (ZWINT-MRPS35P3) and rs7903146 (TCF7L2). Although this work might be of interest, some concerns need to be addressed:
- The authors only pointed out the possible association of rs4462262 (ZWINT-MRPS35P3) and rs7903146 (TCF7L2) with DR, which is not sufficient. In silico functional annotation or experimental validation are required to support the authors' claims. Moreover, possible association between these SNPs and DR pathogenesis should be covered in the Discussion.
Authors’ response: Thanks for pointing this out and your suggestions. As per your suggestions, we have conducted in silico functional analysis for the two SNPs, rs4462262 (ZWINT-MRPS35P3) and rs7903146 (TCF7L2), to assess good amounts of evidence regarding their role in DR pathogenesis. In addition, we have added necessary contents to the methods, results, and discussion sections to reflect our functional annotation and assessment efforts. We believe that these improvements have strengthened the scientific rigor of our study and added significant value to our findings, providing a more comprehensive understanding of the functional relevance of these SNPs in DR pathogenesis.
Methods:
“2.4 Functional annotation of SNPs and gene loci
To understand the functional relevance of the identified SNPs related to DR, we employed in silico functional prediction scores, such as SIFT [46], PolyPhen [47], CADD [48], and RegulomeDB [49]. Moreover, we also assessed SNPs in high (D-prime ≥0.8 and r2 ≥0.8) linkage disequilibrium (LD) with the identified SNPs. The LD data for European populations from the 1000 Genome project was used [50]. Additionally, we analysed the expression quantitative trait loci (eQTL) databases through the Genotype-Tissue Expression (GTEx) portal to gain direct knowledge about effects of risk alleles on nearby gene expression [51].”
Results:
“3.4 Biological relevance of rs4462262 and rs7903146 to DR
SNP rs4462262 is an intergenic variant, located approximately 750 kbp upstream to the nearest gene, IPMK. A microRNA, MIR3924, was situated approximately 110 kbp upstream of the SNP. However, functional annotations for rs4462262 and 54 other SNPs in high LD with rs4462262 were unremarkable (Table 2), providing limited evidence for the biological relevance of these variants.
“In contrast, SNP rs7903146 is located in intron 3 of TCF7L2 and is significantly associated with the expression of TCF7L2 (p=2.9×10-7)(Table 2). Another SNP, rs7074440, in high LD with rs7903146 (D’=0.95 and r2=0.91) has a CADD score of 10.1 and is also significantly associated with the expression of TCF7L2 (p=2.1×10-6)(Table 3), indicating a potentially functional role in TCF7L2 regulation.
Table 2. Functional relevance of rs4462262 and rs7903146 to diabetic retinopathy
SNP |
Position |
Location |
Gene |
SIFT |
PolyPhen |
CADD |
RegulomeDB* |
eQTL (p) |
|
1 |
rs4462262 |
Chr10:59189178 |
Intergenic |
None |
na |
na |
7.10 |
5 |
na |
2 |
rs7903146** |
Chr10:114758349 |
Intronic |
TCF7L2 |
na |
na |
3.27 |
5 |
2.9×10-7 |
3 |
rs7074440 |
Chr10:114785424 |
Intronic |
TCF7L2 |
na |
na |
10.10 |
2b |
2.1×10-6 |
* 2b, TF binding + any motif + DNase Footprint + DNase peak; 5, TF binding or DNase peak. The complete scoring scheme is available at "https://www.regulomedb.org/regulome-help/".
** In complete linkage disequilibrium with rs34872471 identified in GWAS by Xue, Z et al. [20]
CADD, Combined Annotation Dependent Depletion score; eQTL, expression quantitative trait locus; na, not available; PolyPhen, Polymorphism Phenotyping score; RegulomeDB, link: https://cherrylab.stanford.edu/projects/regulomedb; SIFT, Sorting Intolerant From Tolerant score”
Discussion:
“The association between rs7903146 and DR is significant, and the regulatory effects of this SNP and other SNPs in LD provide evidence supporting the functional role of TCF7L2 in DR pathogenesis. As a transcriptional regulator of the canonical Wnt-signaling pathway, TCF7L2 plays a vital role in cell proliferation and fate specification. The involvement of TCF7L2 in DR pathogenesis can be two-fold. First, TCF7L2 may promote pathological retinal neovascularization via ER stress-dependent upregulation of VEGFA, making retinal cells from rs7903146-TT patients more sensitive to VEGF up-regulation and at greater risk of developing PDR [63]. Second, TCF7L2 may also increase risk of DR development and progression by influencing the responses to glycemic control drugs, such as glipizide and metformin which are two commonly used glycemic control drugs [71]. And good glycemic control is closely associated with short- and long-term beneficial effects on DR prevention and intervention [72,73]. However, the functional relevance of rare mutations in the TCF7L2 gene and their impact on DR pathophysiology in diverse ethnic groups remains unknown, and further sequencing studies of large cohorts are needed to understand their role. In addition, understanding the functional regions of the TCF7L2 gene and their impact on specific phenotypes will also be important to comprehend their role in diverse human diseases, including DR [74,75].
“On the other hand, the lack of functional annotations for rs4462262 and 54 other SNPs in high LD with rs4462262 limits our understanding of their functional relevance in DR pathogenesis. To reveal their role in DR, identification of rare and common variants that truly alter the risk DR and functional investigations are crucial. Confirmation and functional annotation of these variants could be a critical step in discovering new genetic markers for DR, facilitating better prediction of disease susceptibility and tailoring individualized treatments for patients.”
- The first two paragraphs of the Introduction both presented general knowledge of DR, which can be merged into one paragraph.
Authors’ response: Thanks for the suggestion. We have combined these two paragraphs.
“Diabetic retinopathy (DR) is a leading cause of visual impairment. It leads to blinding complications and affects more than one third of global diabetic patients posing blindness to one in three DR patients [1]. DR is a progressive retinal microvascular complication of diabetes mellitus. Its pathology involves microaneurysms, haemostatic changes, and capillary occlusion in the retina. The vessel permeability increases, resulting in retinal non-perfusion, neovascularization and vessel wall thickening. Further progress into macular edema, vitreous hemorrhage or retina distortion can lead to severe visual loss and blindness [2]. Effective interventions, such as tight glycemic and blood pressure control, lipid-lowering, and laser therapies, could delay its progression and preserve vision. Early diagnosis and appropriate treatment can help prevent visual damage and the blinding consequences [3-8].”
- The authors should highlight the significance and novelty of their work in the last paragraph of the Introduction and the Discussion.
Authors’ response: Thanks for the suggestion. We have re-wrote part of the introduction and discussion sections to highlight the significance and novelty of our work.
Introduction: “Our study provides a comprehensive summary and update, emphasizing the urgent need for improved consensus among DR GWASs. Through meta-analysis of reported GWAS SNPs, we confirmed the association of two SNPs, rs4462262 (ZWINT-MRPS35P3) and rs7903146 (TCF7L2), with DR. However, to ensure the robustness and generalizability of GWAS findings, further replication of the candidate SNPs summarized in our study is necessary.”
Discussion/conclusion: “Our study underlines the urgent need for consistent replication efforts in the field of DR genetics. Two SNPs, rs4462262 (ZWINT-MRPS35P3) and rs7903146 (TCF7L2), showed consistent association with DR. Confirmation and functional investigations of reported and new variants are needed for a better understanding of the underlying mechanisms of DR development.”
- The hyphens in line 14 and 16 should be removed.
Authors’ response: Sorry for these formatting errors. We have corrected them and checked the rest of the texts.
“The present study aimed to summarize and validate the genomic association signals for diabetic retinopathy (DR), proliferative DR, and diabetic macular edema/diabetic maculopathy. A systematic search of the GWAS catalog and PubMed/MELINE databases was conducted to curate a comprehensive list of significant GWAS discoveries. The top signals were then subjected to meta-analysis using established protocols. The results indicate the need for improved consensus among DR GWASs, highlighting the importance of validation efforts. Subsequent meta-analysis confirmed the association of two SNPs, rs4462262 (ZWINT-MRPS35P3)(odds ratio=1.38, p=0.001) and rs7903146 (TCF7L2)(odd ratio=1.30, P<0.001), with DR in independent populations, strengthening the evidence of their true association. We also compiled a list of candidate SNPs for further validation. This study highlights the importance of consistent validation and replication efforts in the field of DR genetics. The two identified gene locus warrant further functional investigation to understand their role in DR pathogenesis.”
- The column name “PMID” in Table 1 should be corrected.
Authors’ response: Thanks for pointing this out. We have changed it to ‘Reference.’

Reviewer 2 Report
In this paper, the authors summarized the results from published GWASs on DR, PDR, and DME. The meta-analysis of these GWAS indicated the need for improved consensus and validation among the studies. They also confirmed 2 SNPs associated with DR. The methods are well described and results are appropriately interpreted. However, the significance of this study was not clearly defined. The discussion part has room for improvement to incorporate the interpretation of the GWAS meta-analysis of PDR and DME and highlight the importance of this study. More detail on the confirmed SNPs will strengthen the discussion.
Author Response
Dear Reviewer,
Thank you for your favorable comments and valuable suggestions regarding our manuscript. Your input has been instrumental in strengthening the quality and impact of our study.
As per your recommendations, we have made changes to the introduction and discussion sections to emphasize the significance of our research. We have also included new results from the functional analysis of the two SNPs, along with new discussions about the mechanisms underlying the associations of these SNPs and relevant genes to DR pathogenesis. We believe that these improvements have significantly enhanced the scientific rigor of our study, providing a more comprehensive understanding of the functional relevance of the identified SNPs in DR pathogenesis.
We have highlighted all the changes we made in yellow, to make it easier for you to identify the revisions. We hope that these changes address all of your concerns, and we remain open to any further suggestions or feedback you may have.
Once again, thank you for your valuable input and your time. We are grateful for your support and guidance, and we look forward to your feedback on the revised manuscript.
Sincerely,
Shisong Rong
Department of Ophthalmology, Massachusetts Eye and Ear,
Mass General Brigham, Harvard Medical School,
Boston, MA, USA
In this paper, the authors summarized the results from published GWASs on DR, PDR, and DME. The meta-analysis of these GWAS indicated the need for improved consensus and validation among the studies. They also confirmed 2 SNPs associated with DR. The methods are well described and results are appropriately interpreted. However, the significance of this study was not clearly defined. The discussion part has room for improvement to incorporate the interpretation of the GWAS meta-analysis of PDR and DME and highlight the importance of this study. More detail on the confirmed SNPs will strengthen the discussion.
Authors’ responses:
Thank you for your favorable comments and valuable suggestions regarding our manuscript. Your input has been instrumental in strengthening the quality and impact of our study.
As per your recommendations, we have made changes to the introduction and discussion sections to emphasize the significance of our research. We have also included new results from the functional analysis of the two SNPs, along with new discussions about the mechanisms underlying the associations of these SNPs and relevant genes to DR pathogenesis. We believe that these improvements have significantly enhanced the scientific rigor of our study, providing a more comprehensive understanding of the functional relevance of the identified SNPs in DR pathogenesis.
We have highlighted all the changes we made in yellow, to make it easier for you to identify the revisions. We hope that these changes address all of your concerns, and we remain open to any further suggestions or feedback you may have.
Once again, thank you for your valuable input and your time. We are grateful for your support and guidance, and we look forward to your feedback on the revised manuscript.
[Significance of this study]
Introduction: “Our study provides a comprehensive summary and update, emphasizing the urgent need for improved consensus among DR GWASs. Through meta-analysis of reported GWAS SNPs, we confirmed the association of two SNPs, rs4462262 (ZWINT-MRPS35P3) and rs7903146 (TCF7L2), with DR. However, to ensure the robustness and generalizability of GWAS findings, further replication of the candidate SNPs summarized in our study is necessary.”
Discussion/conclusion: “Our study underlines the urgent need for consistent replication efforts in the field of DR genetics. Two SNPs, rs4462262 (ZWINT-MRPS35P3) and rs7903146 (TCF7L2), showed consistent association with DR. Confirmation and functional investigations of reported and new variants are needed for a better understanding of the underlying mechanisms of DR development.”
[Functional analysis of identified variants and related gene/s]
Methods:
“2.4 Functional annotation of SNPs and gene loci
To understand the functional relevance of the identified SNPs related to DR, we employed in silico functional prediction scores, such as SIFT [46], PolyPhen [47], CADD [48], and RegulomeDB [49]. Moreover, we also assessed SNPs in high (D-prime ≥0.8 and r2 ≥0.8) linkage disequilibrium (LD) with the identified SNPs. The LD data for European populations from the 1000 Genome project was used [50]. Additionally, we analysed the expression quantitative trait loci (eQTL) databases through the Genotype-Tissue Expression (GTEx) portal to gain direct knowledge about effects of risk alleles on nearby gene expression [51].”
Results:
“3.4 Biological relevance of rs4462262 and rs7903146 to DR
SNP rs4462262 is an intergenic variant, located approximately 750 kbp upstream to the nearest gene, IPMK. A microRNA, MIR3924, was situated approximately 110 kbp upstream of the SNP. However, functional annotations for rs4462262 and 54 other SNPs in high LD with rs4462262 were unremarkable (Table 2), providing limited evidence for the biological relevance of these variants.
“In contrast, SNP rs7903146 is located in intron 3 of TCF7L2 and is significantly associated with the expression of TCF7L2 (p=2.9×10-7)(Table 2). Another SNP, rs7074440, in high LD with rs7903146 (D’=0.95 and r2=0.91) has a CADD score of 10.1 and is also significantly associated with the expression of TCF7L2 (p=2.1×10-6)(Table 3), indicating a potentially functional role in TCF7L2 regulation.
Table 2. Functional relevance of rs4462262 and rs7903146 to diabetic retinopathy
SNP |
Position |
Location |
Gene |
SIFT |
PolyPhen |
CADD |
RegulomeDB* |
eQTL (p) |
|
1 |
rs4462262 |
Chr10:59189178 |
Intergenic |
None |
na |
na |
7.10 |
5 |
na |
2 |
rs7903146** |
Chr10:114758349 |
Intronic |
TCF7L2 |
na |
na |
3.27 |
5 |
2.90E-07 |
3 |
rs7074440 |
Chr10:114785424 |
Intronic |
TCF7L2 |
na |
na |
10.10 |
2b |
2.10E-06 |
* 2b, TF binding + any motif + DNase Footprint + DNase peak; 5, TF binding or DNase peak. The complete scoring scheme is available at "https://www.regulomedb.org/regulome-help/".
** In complete linkage disequilibrium with rs34872471 identified in GWAS by Xue, Z et al. [20]
CADD, Combined Annotation Dependent Depletion score; eQTL, expression quantitative trait locus; na, not available; PolyPhen, Polymorphism Phenotyping score; RegulomeDB, link: https://cherrylab.stanford.edu/projects/regulomedb; SIFT, Sorting Intolerant From Tolerant score”
Discussion:
“The association between rs7903146 and DR is significant, and the regulatory effects of this SNP and other SNPs in LD provide evidence supporting the functional role of TCF7L2 in DR pathogenesis. As a transcriptional regulator of the canonical Wnt-signaling pathway, TCF7L2 plays a vital role in cell proliferation and fate specification. The involvement of TCF7L2 in DR pathogenesis can be two-fold. First, TCF7L2 may promote pathological retinal neovascularization via ER stress-dependent upregulation of VEGFA, making retinal cells from rs7903146-TT patients more sensitive to VEGF up-regulation and at greater risk of developing PDR [63]. Second, TCF7L2 may also increase risk of DR development and progression by influencing the responses to glycemic control drugs, such as glipizide and metformin which are two commonly used glycemic control drugs [71]. And good glycemic control is closely associated with short- and long-term beneficial effects on DR prevention and intervention [72,73]. However, the functional relevance of rare mutations in the TCF7L2 gene and their impact on DR pathophysiology in diverse ethnic groups remains unknown, and further sequencing studies of large cohorts are needed to understand their role. In addition, understanding the functional regions of the TCF7L2 gene and their impact on specific phenotypes will also be important to comprehend their role in diverse human diseases, including DR [74,75].
“On the other hand, the lack of functional annotations for rs4462262 and 54 other SNPs in high LD with rs4462262 limits our understanding of their functional relevance in DR pathogenesis. To reveal their role in DR, identification of rare and common variants that truly alter the risk DR and functional investigations are crucial. Confirmation and functional annotation of these variants could be a critical step in discovering new genetic markers for DR, facilitating better prediction of disease susceptibility and tailoring individualized treatments for patients.”
